# Size-Exclusion Chromatography as a Technique for the Investigation of Novel Extracellular Vesicles in Cancer

**DOI:** 10.3390/cancers12113156

**Published:** 2020-10-27

**Authors:** Daniel S. K. Liu, Flora M. Upton, Eleanor Rees, Christopher Limb, Long R. Jiao, Jonathan Krell, Adam E. Frampton

**Affiliations:** 1Department of Surgery & Cancer, Imperial College London, Hammersmith Hospital Campus, Du Cane Road, London W12 0HS, UK; daniel.liu08@imperial.ac.uk (D.S.K.L.); f.upton19@imperial.ac.uk (F.M.U.); e.rees@imperial.ac.uk (E.R.); j.krell@imperial.ac.uk (J.K.); 2Hepato-Pancreato-Biliary (HPB) Surgical Unit, Department of Surgery & Cancer, Imperial College London, Hammersmith Hospital Campus, Du Cane Road, London W12 0HS, UK; l.jiao@imperial.ac.uk; 3Hepato-Pancreato-Biliary (HPB) Surgical Unit, Royal Surrey County Hospital NHS Foundation Trust, Guildford, Surrey GU2 7XX, UK; c.limb@surrey.ac.uk; 4Department of Clinical and Experimental Medicine, Faculty of Health and Medical Sciences, The Leggett Building, University of Surrey, Guildford, Surrey GU2 7WG, UK

**Keywords:** size-exclusion chromatography, SEC, exosomes, extracellular vesicles, cancer, biomarkers

## Abstract

**Simple Summary:**

Extracellular vesicles (EVs) are small particles that are released by cancer cells, and they may hold vital information for researchers looking for early markers for diagnosis. Size-exclusion chromatography (SEC) is a classical technique that has become increasingly popular and can be used for rapid isolation and investigation of both their cargo and functionality. This systematic review highlights its main technical aspects, the type of materials involved and by covering the findings of the identified papers hopes to demonstrate the utility of this method in cancer research to date.

**Abstract:**

Cancer cells release extracellular vesicles, which are a rich target for biomarker discovery and provide a promising mechanism for liquid biopsy. Size-exclusion chromatography (SEC) is an increasingly popular technique, which has been rediscovered for the purposes of extracellular vesicle (EV) isolation and purification from diverse biofluids. A systematic review was undertaken to identify all papers that described size exclusion as their primary EV isolation method in cancer research. In all, 37 papers were identified and discussed, which showcases the breadth of applications in which EVs can be utilised, from proteomics, to RNA, and through to functionality. A range of different methods are highlighted, with Sepharose-based techniques predominating. EVs isolated using SEC are able to identify cancer cells, highlight active pathways in tumourigenesis, clinically distinguish cohorts, and remain functionally active for further experiments.

## 1. Introduction

Extracellular vesicles (EVs) are small particles defined by a lipid bilayer and an inability to replicate, and they lack organelles or a functional nucleus. Historically, they have been termed various names, such as “shedding membrane vesicles”, “membrane fragments” [1], “microvesicles” [2], or more recently “exosomes”. We now make a broad distinction between subsets of EVs by their biogenesis, size, or density, and this is dependent on the experimental isolation method used.

Whilst larger structures (200–1000 nm) tend to be plasma membrane-derived apoptotic bodies and microvesicles [3], smaller structures (30–150 nm), which are of endocytic origin (released into the extracellular environment from fusion of multivesicular bodies (MVB) with the plasma membrane), are termed exosomes [4]. However, there remains a considerable overlap in what is isolated for experimental characterisation, and studies investigating vesicular biogenesis show that both large and small vesicles can demonstrate direct budding from the plasma membrane [5]. Exosomes are the dominant area of research, having been shown to act as a distinct intercellular signalling mechanism, shuttling messenger RNAs (mRNAs) and microRNAs (miRNAs) with the potential to regulate functional expression in other cells [6]. When there is no clear demonstration of endosomal origin, the International Society for Extracellular Vesicles (ISEV) [7,8] suggests use of the term EVs to improve clarity.

The release of all types of EVs by tumour cells is of particular interest to researchers investigating potential biomarkers, as these may be a source of several targets in the growing field of “liquid biopsy” for cancers that are difficult to access or diagnose. This idea involves the investigation of circulating tumour cells, free nucleic acids, and proteins, which are detectable in blood and biofluids, including from samples as diverse as breast milk [9], saliva [10], ascites [11], and bile [12].

### 1.1. Isolation Methods in Brief

There are several methods for the isolation and enrichment of EVs from both cell-conditioned media (CCM) and biofluids. Historically, differential ultracentrifugation (dUC) has been popular, as it uses commonly available laboratory facilities, with low costs and simple protocols to obtain pelleted EVs. In 2016, a survey showed that 81% of all experiments registered used dUC as their primary EV isolation method [13]. First, low speeds are used to remove cellular debris and subcellular organelles, then protocols using speeds of approximately 100,000 × *g* for 1–2 h are utilised to retrieve EV pellets. Large volumes of cell-conditioned medium can be processed, and the use of a washing step and repeat spin, as well as filtration, can further enhance the purity of the EVs and remove any contaminating soluble proteins.

This has given way to several other techniques, designed to exploit the different characteristics of EVs, including by size, density, and known surface markers. By 2017, dUC accounted for 45% of all experiments with a reported 190 unique isolation methods registered on EV-TRACK, an online tool designed to improve standardisation across the field [14]. The choice of method is fundamentally based on a trade-off between EV yield and purity from contaminants such as albumin that may be highly represented in samples [15]. Differential ultracentrifugation separates based on a sedimentation coefficient, which is in turn determined by the mass of the particle and the distance travelled. Therefore, small changes in centrifugation protocols are likely to lead to a wide variation in pelleted material [16].

Exosomes have been found to float at specific densities ranging from 1.05 to 1.25 g/mL on continuous sucrose gradients [17], and this has been used as a method for separation of a highly pure EV fraction. This results in a relatively low yield and remains labour intensive as well as requiring eventual separation from the gradient buffer made of iodixanol or sucrose.

As well as separation by density and/or size, immunoaffinity using biotin-labelled antibody capture beads [18] can be undertaken as well as polyethylene glycol (PEG) precipitation as commonly used in many commercial kits (ExoQuick by System Biosciences, Palo Alto, CA, USA, is one example). Whilst the former is highly selective and, therefore, has a low overall yield, the latter is both rapid and technically easy but co-isolates non-vesicular contaminants to a greater degree [19,20].

Size-exclusion chromatography (SEC) was shown in 2016 to account for 5.6% of experiments but has become more popular as interest in EV isolation methods with diagnostic potential has accelerated in the past four years. In this regard, the advantage of a single-step technique able to isolate from small sample sizes may be of great interest.

### 1.2. Size-Exclusion Chromatography, an Old but New Technique

Size-exclusion chromatography (SEC) was initially designed as a technique to fractionate amino acids and protein fragments by molecular weight and first utilised starch in a process termed “partition chromatography” [21].

The process of chromatography involves the use of a stationary phase (in this case a gel; see Figure 1) held in a column, which allows a liquid mobile phase to pass through. This contains the analyte in aqueous buffer solution, which leaves the column at a rate proportional to its size, or more accurately the hydrodynamic volume [19]. With complex mixtures made up of different-sized particles, larger molecules are excluded from the gel and are recovered quickly, whilst smaller particles are impeded and elute much later.

The initial experiments by researchers working at Queen Charlotte’s Hospital in London used potato and maize starch and were noted to be inconsistent in size and behaviour leading to different elution volumes in each column [22]. This was improved by the subsequent development of dextran and crosslinked agarose, which is a polysaccharide polymer material extracted from seaweed. Today, gel matrices are known by tradenames and include dextran polymers (Sephadex (Cytiva, Marlborough, MA, USA)), agarose (Sepharose (Cytiva, Marlborough, MA, USA)), or polyacrylamide (Sephacryl (Cytiva, Marlborough, MA, USA) or BioGel P (Bio-Rad, Hercules, CA, USA)) manufactured as beads and optimised for experimental consistency.

The transit time of an analyte must be experimentally determined and is roughly based on the size of the pores within the gel matrix. Additionally, different gels have an optimal molecular weight range, which will allow good linear separation, and it is recommended that this is usually calibrated with a standard curve based on known weights. Other factors that will affect the process are the dimensions of the column, usage of column frits and consistency of the process of forming the gel within the column, which can also be minimised by use of industrially standardised columns on the market. Larger bed volumes can lead to an improved resolution of larger particles, but this must be balanced by a longer time taken to elute each fraction [23].

Literature on SEC as a method for EV isolation has become more prevalent, with this current review identifying more papers year-on-year. The benefits of SEC are that it is rapid, taking 10 to 20 min per sample; relatively inexpensive; and able to efficiently remove over 95% of blood protein, which makes it extremely clinically applicable when compared with dUC or other techniques [24].

The difficulties inherent in the process include preparatory steps to remove large contaminants, the possible co-elution of similar sized lipoproteins, and any concentration steps that may be required either before or after. Lipoproteins such as very-low-density lipoprotein (VLDL) have a similar size to EVs and are a recognised co-precipitant from blood plasma or serum samples, which should be identified by the presence of apolipoproteins and a lipid monolayer [25]. This is an important factor to consider, as lipoproteins are carriers of potentially functional miRNAs and, so, could complicate potential biomarker discovery [26]. Combining SEC with other methods, such as immunoprecipitation is possible and has been demonstrated in several papers within this review. Further work may also be needed to determine the interaction between EVs and gels commonly in use. EVs have a large surface area to volume ratio for interaction, and this is a potential area of future research [27].

Uses of SEC within the EV field also include separation of vesicles from free dye such as removing PKH or CFSE (CarboxyFluorescein diacetate Succinimidyl Ester) from treated EVs prior to subsequent analysis [28], as well as separating EVs from iodixanol used in gradient density methods [29]. This use of orthogonal approaches, i.e., both size and density purification in EV isolation and purification, is increasing as protocols are shared and familiarity with the different methods improves.

## 2. Methodology of Review

### 2.1. Objectives

The overall aim of the review was to explore the use of size-exclusion chromatography (SEC) (also known as gel filtration) systematically in the context of EV isolation and its use in the cancer biomarker discovery field.

### 2.2. Search Methods

Databases searched included MEDLINE (https://www.nlm.nih.gov/bsd/medline.html), EMBASE (https://www.embase.com/), and Journals@Ovid using Ovid SP (https://www.ovid.com/; accessed last on 20 April 2020; see Appendix B for search terms). Additional papers were identified by iteratively searching through reference lists and previous work by the same authors. Finally, a search of EV-TRACK for all experiments using “size-exclusion chromatography” as a separation protocol was undertaken, and the relevant papers were included.

### 2.3. Selection Criteria

The focus of this review is on original studies that used SEC, or gel filtration, as a primary method of EV isolation in either a cancer cell line or clinical sample from a cancer cohort. After excluding duplicates, abstracts were reviewed and we excluded any descriptive protocol articles, conference abstracts, or non-original research. Forty full-text articles were reviewed, and those found to use SEC as an adjunct method (e.g., removal of dye from pre-isolated EVs) were also excluded (see Appendix A).

## 3. Results

### 3.1. Comparison of Isolation Methods

We looked at 37 studies, of which there were 39 different isolation methods described (shown in Table 1).

Sephacryl and Sepharose were two of the most common polymers identified in the papers examined. Sepharose appeared to be the most popular, with 2B and 4B referring to the percentage of crosslinked agarose (2–4%). MiniSEC, the method that was coined and used solely by the Whiteside lab group, uses small-size ion exchange columns and has resulted in a significant number of publications.

Proprietary methods include the qEV column introduced in 2015 by Izon Science (Christchurch, New Zealand), and the EVSecond (GL Science, Tokyo, Japan). One of the theoretical benefits of industrially manufactured columns is reduced operator dependence in the making of Sepharose columns, leading to better uniformity. The qEV columns come with ISO 13485:2016 compliance, which requires that the certified organisation demonstrates a quality management system to maintain certain standards and systems during the manufacturing and delivery process. Although this does not guarantee that EVs are isolated, it is important that elution patterns are characterised for all types of columns prior to use.

### 3.2. EV Characterisation, Concordance with MISEV Guidelines

Minimal Information for Studies of Extracellular Vesicles (MISEV2018) [8] was an updated version of guidelines developed in 2014 [7] to sensitise researchers to the experimental and reporting requirements in the EV field. By standardising nomenclature and experimental methods, it is hoped that this will improve the reliability and reproducibility of studies.

EV-TRACK is an open-source knowledgebase which was endorsed by MISEV2018 and uses a key feature, termed an EV-METRIC (taken from http://www.evtrack.org/about.php) to highlight nine specific areas that can be used to support critical appraisal of an EV study (Table 2) [30].

Appendix A show the isolation methods identified and the characterisation used, which is a vital part of calculating the EV-METRIC. Whilst centrifugation specifics have been noted, density experiments were only undertaken to assess SEC-isolated EVs in four studies [31,32,33]. We have therefore focused our assessment of the quality of studies using the EV-METRIC, specifically the analysis of proteins and particle characteristics (i.e., size, distribution, concentration, morphology), in the hope that this will stimulate discussion of the differences between the studies.

Protein analysis whether by Western blot, flow cytometry or mass spectroscopy for EV-enriched proteins is a feature of nearly every study reviewed (84% of studies), and Western blots for the main tetraspanins, such as CD63, CD9, and CD81 have become a surrogate for the demonstration of EVs, alongside electron microscopy (EM) and nanoparticle tracking analysis (NTA). Whilst these have been established for some time [34], it is worth bearing in mind that they can be enriched in vesicles that are formed in both an endosomal and non-endosomal fashion (i.e., not just exosomal) [35]. Alix (assessed in 11% of studies), and tumour susceptibility gene 101 (TSG101; noted in 41% of studies), which both relate to initial intraluminal vesicle (ILV) formation in the endosomal sorting complex required for transport (ESCRT) machinery, are used as common markers of EVs. There is also significant heterogeneity when it comes to clinical samples, and it is not expected that all markers are constitutively expressed in EVs. Furthermore, only three studies investigated non-EV markers, which may include lipoproteins and serum-derived materials, or proteins derived from subcellular structures not present in EVs. Examples of this might be proteins derived from the endoplasmic reticulum (such as grp94 or calnexin) or mitochondrial proteins such as cytochrome C.

### 3.3. EV-Associated Cargo

EVs have been found to contain a diverse array of cargo, which includes lipids, soluble and transmembrane proteins, RNAs, and DNA. Additionally, some studies have focussed on the function of EVs and their effect on recipient cells. In order to analyse the findings, we have divided the studies by the general -omics approach they have used (see Table 3). This split 37 studies into those looking at functionality of EVs; proteomics; feasibility studies; and a small number investigating the RNA content of isolated EVs.

#### 3.3.1. EV Proteomics

EVs are associated with pro-apoptotic protein markers and stimulate an immune response that may promote the tumour microenvironment

The study of protein cargo from EVs has been a key area of research, with studies investigating the contents of “shedding membrane vesicles” obtained by gel filtration dating back to 2003 [36]. Initial studies looked into Fas ligand (FasL), and its immunosuppressive effects on T-cells, an area of significant interest in the 1990s, as EVs had been shown to demonstrate antigen-presenting ability to T-cells [37]. It had been known for some time that cancer cells produced a large number of EVs, and that these were detectable in patients with cancer [38]. However, it was unclear whether they were the cause or effect of increases in synthetic, metabolic, or catabolic pathways [39].

Taylor et al. [36] is the earliest study covered by our review and showed that the Fas/FasL pathway plays a dual role in vivo, mediating proinflammatory effects, as well as immune cell apoptosis. The 42 kDa cell-membrane-associated protein was detected in these vesicles and led to the apoptosis of activated normal T lymphocytes ex vivo. These data were supported in both oral and ovarian cancer cohorts [2].

Since those early investigations, EVs have been found to play a much wider role to include cell signalling and metabolism. This includes studies such as Hong et al. [40] and Kawakami et al. [31], which focussed on the presence of proteins such as TGF-β1 (Transforming growth factor beta 1) and GGT (Gamma-glutamyltransferase). TGF-β1 can be carried by exosomes and be found to be raised in patients with acute myeloid leukemia (AML) but normalised after long-term complete remission. This was better correlated with clinical disease status than broad measurements of total exosomal protein levels. However, they were unable to determine a subset of EVs, which were responsible for TGF-β1 carriage.

GGT, the activity of which is upregulated in colon, ovary, and liver cancer; astrocytic glioma; soft tissue sarcoma; melanoma; and leukaemia, was drawn from a selection of highly upregulated proteins identified from an androgen-resistant cell line and found to be a serum biomarker raised in exosomes that could distinguish patients with prostate cancer from those with benign prostatic hypertrophy (BPH). However, they used multiple methodologies for their isolation, alternating between dUC and SEC, as well as density gradient and immunocapture. Although EVs are obtainable by all these methods, in multiple comparative studies, these appear to isolate EV populations with different size distributions and protein content, which suggests that the use of a consistent isolation methodology is best practice [41,42].

EV research has begun to talk less of shedding (i.e., a passive excretory mechanism for EVs), and instead has focussed on EVs as perhaps being actively released as part of cancer progression. One study that looked into this was reported by Djusberg et al. [43] and showed that overexpression of YIPF6 (a protein co-expressed with androgen receptors) in 22R v1 prostate cancer cells resulted in enhanced secretion of EVs.

The report by Broggi et al. [44] used K14-VEGFR-3-Ig transgenic mice lacking dermal lymphatics, to show that lymphatics were a major route of EV trafficking from the interstitial space into the lymph fluid, lymph nodes (LNs), and systemic circulation, and that this could be a source of tumour-derived EVs.

However, this study suffered from methodological flaws, using dUC to investigate miRNAs, and switching to an in vivo study using SEC. It is possible that although their clinical study detected metastasis-related miRNAs specifically expressed in the EVs of metastatic melanoma patients, they were unable to detect a statistical difference between cohorts since they investigated EVs with lower purity. Another criticism is the use of ultracentrifugation supernatant as a “non-EV” comparator. Indeed, it cannot be assumed that supernatant generated after UC is a suitable non-EV sample for comparison, as it has been shown that single spins of 1–2 h may only remove 40–50% of EVs [45].

##### EV Protein Markers Are Differentially Expressed by Cancer Cells

SEC can be used to effectively isolate EVs from conditioned cell culture medium, and this has led to the development of several specific EV-associated markers that could be implicated in tumourigenesis.

Theodoraki et al. [46,47,48] have written three papers looking at patients with head and neck cancer (HNSCC) using a process termed “immune affinity capture” to magnetically label specific subsets of EVs isolated by SEC with antibodies. This technique enables downstream analysis of EVs that express specific protein markers and, therefore, may be attributed to T-cells or otherwise. T-cell specific EVs were isolated based on CD3+ expression and found to constitute between 38% and 54% of total exosomes isolated from plasma samples, a higher proportion than that found in a healthy cohort. Conversely, CD3− EVs (potentially tumour-derived) were found to contain high levels of CD44v 3 protein and to be associated with unfavourable clinicopathological parameters.

They also focussed on the PD-1/PD-L1 (Programmed death-ligand 1) immunosuppression pathway, with high PD-L1 expression in tumours being associated with poor prognosis. They were able to show expression of PD-L1 on EVs was associated with more disease activity and UICC (Union for International Cancer Control) staging, as well as activity on normal T-cells in vitro.

Interestingly, they measured both surface PD-L1 cargo in exosomes and PD-L1 levels in patient plasma but did not find a correlation. This may indicate that exosomal PD-L1 expression is an independent biomarker distinct from soluble protein levels, with proven activity on peripheral cells. Unfortunately, tissue correlation was never investigated. It would have also been especially interesting to see whether this activity was modulated by PD-1 inhibitors, such as pembrolizumab, although these experiments have subsequently been shown with EVs isolated via UC in melanoma cells [49], and appear to be a promising area for future research [50].

Stress-induced pathways, such as p53 activation (non-mutant), lead to increased EV secretion via *TSAP6* and *CHMP4C* gene transcription [51,52]. Differential EV expression in cell injury caused by ionising radiation (IR) may therefore be a good model to investigate EVs as biomarkers during cancer treatment. Abramowicz et al. [53] showed that in a p53-wildtype Human Papillomavirus (HPV)- associated head and neck cancer cell line, EVs released by irradiated cells had increased levels of proteins, such as CD63 and CD81, whereas other surface protein levels remained unchanged (i.e., CD9 and TSG101). Gene ontology analyses showed associations between upregulated proteins and DNA repair, regulation of reactive oxygen species metabolism, and de novo protein folding. Unusually, histones and chromatin-associated factors were also identified, which may be related to the increased loading of exosomal DNA during cellular senescence [54].

Freitas et al. [55] demonstrated that altering the glycosylation profile of gastric adenocarcinoma cells, MKN45, also leads to a change in EV glycosylation. Comparing well-known isolation methods, they found that only 16.3% of proteins were common to all EV isolation methods, with SEC and Optiprep Density Gradients (ODG) resulting in EVs with the highest concentration of EV markers. Interestingly, the EVs derived were particularly enriched for specific glycoproteins carrying the Sialyl-Tn antigen (STn), compared with cell lysate. Synthesis of STn is associated with poor prognosis in gastric cancer and is a widely expressed epithelial cancer marker [56].

Furthermore, SEC can also be used to distinguish the proteome of EVs from cell lines representing HPV+ and HPV− head and neck cancer. Ludwig et al. [57] showed differential protein expression, particularly with upregulation of CD47 in HPV+ HNSCC cells, which inhibited exosome uptake/phagocytosis by human monocytes, and may play a role in the more robust, virus-primed anti-tumour immunity present in HPV+ malignancies [58].

##### EV Expression Highlights Differences in Clinical Outcome and Surgical Resectability

The recent paper by Theodoraki et al. [48] demonstrated how SEC can be a highly sensitive method to accurately determine EV protein expression. In this study, they developed a custom antibody microarray to further characterise the EV surface antigen expression in blood plasma from a cohort of HNSCC patients in a longitudinal manner, before and after treatment with cetuximab, radiotherapy, and ipilimumab. They then followed these patients up for 24 months to assess EVs as a predictive marker for disease recurrence.

This microarray consisted of classical exosome surface markers such as CD9, CD63, CD81, and tumour-enriched exosome “TEX” markers, which included Epidermal Growth Factor Receptor 1 (EGFR1), Melanoma-associated antigen 3 (MAGEA3), Epithelial cell adhesion molecule (EpCAM), and Chondroitin sulfate proteoglycan 4 (CSPG4). TEX markers were not detected in their small cohort of healthy controls (*n* = 5), whilst in the cancer patients, the TEX/Total exosome ratio, as well as total exosomal protein levels, were significantly different in patients who had recurrence after treatment. In assessing subsets of EVs, they returned to the immunoaffinity capture method to look at CD3, PD-L1, and CTLA4 (cytotoxic T-lymphocyte-associated protein 4) expression on T-cell derived EVs, and highlighted that high levels of CD3-/PD-L1+ and CD3-/CTLA4+ exosomes at baseline might predict patients who would benefit from immunotherapy.

EVs from SEC can also correlate with a patient’s tumour grade and the extent of surgical removal, in patients presenting with brain glioma [59]. Liquid chromatography mass spectroscopy (LC-MS) identified 12 plasma EV-associated proteins that were significantly different between patients with glioblastoma and low-grade glioma. The authors were especially interested in Syndecan-1, a membrane protein, which is now known to interact with the well-known EV markers Syntenin-ALIX, which facilitate membrane budding [60]. Syndecan-1 had an exceptionally high AUC (Area Under the Curve on the receiver operating characteristic) value of 0.81 and a sensitivity and specificity of 71% and 91%, respectively, for distinguishing these high- and low-grade tumours.

However, in both these studies, they did not compare EV concentration between the cohorts, which could account for the increase in protein expression between disease groups. Normalisation by either EV number, or an established EV marker, would improve the detection of differentially expressed EV-associated proteins.

Nevertheless, it appears that there are EV-associated proteins increased in cancer, and these are readily detectable in small cohort clinical trials using SEC isolation. The majority of proteins involved in transformation to cancer are usually downregulated [61], so focussing on a subset that cause EV release may be promising. With the dynamic range of blood plasma protein concentrations varying by >10 orders of magnitude [62], these studies show overall that SEC is an efficient technique for the removal of a high abundance protein, thus allowing for greater sensitivity and detection of specific proteins secreted by cancer cells.

#### 3.3.2. EV RNA

##### Clinical Studies Show That EVs from SEC Are Enriched with Functionally Active Small RNAs

A small number of studies (*n* = 4; see Appendix A) have investigated RNA expression within EVs after using SEC as an isolation method. Two were clinical studies, whilst the other two were in vitro studies [63,64].

The first study by Rabinowits et al. [65], (published two years after the paper by Valadi et al. highlighting the role of EVs in transferring functional mRNAs and miRNAs [6]) used EpCAM immunoaffinity capture to purify for EVs present in the plasma of patients with lung cancer. EpCAM has been shown to be ubiquitously expressed on epithelial tumours [66] and on EVs from these cancers [67,68]. A microarray containing probes for 467 human mature miRNAs was used to rapidly identify potential miRNA candidates. As has been demonstrated in other papers [69], EV concentration was higher in the cancer cohort, with a higher RNA concentration. Interestingly total exosome and total miRNA levels were also significantly higher in their cancer cohort, but the paper did not report on differentially expressed miRNA candidates.

Van Eijndhoven et al. [33] investigated EVs obtained from patients with classical Hodgkin’s lymphoma (cHL), comparing these patients with healthy controls and patients with Systemic Lupus Erythematosus. This study had several advantages. Firstly, the authors used RNA sequencing to compare EVs from the clinical cohorts, and secondly, utilising SEC they were able to compare EV fractions with RNA extracted from the protein/High-Density Lipoprotein (HDL) fraction generated from the same samples. This protein/HDL fraction contained a significant number of miRNA reads but at a relatively low complexity (<120 unique species), compared with the comprehensive miRNA repertoires of cHL patients obtained from EVs, with miR-486-5p and miR-92a-3p being highly abundant. They were then able to identify a panel of five cHL-related miRNAs (i.e., miR-127-3p, miR-155-5p, miR-21-5p, let-7a-5p). Next, using RT-qPCR validation of miR-127-3p and miR-155-5p, they were able to show that miRNA fold differences were more robust in EVs, compared with those seen in total plasma. Additionally, in a small cohort (*n* = 7) that was followed longitudinally, levels of miR-127-3p, miR-155-5p, and miR-21-5p in EVs decreased markedly after treatment, whilst levels of these miRNAs in total plasma did not.

The major limitation of this study was the lack of assessment for EV-associated protein markers, which should be included if we want to conclude that this is truly an EV study as per ISEV guidelines. Instead, PKH67-labelled (a type of lipophilic dye) flow cytometry was used to show evidence of EVs. This is generally not recommended as non-EV species, such as lipoproteins, are not distinguished separately [70,71].

##### EVs from Cell Lines Can also Be a Useful Source of RNA-Based Biomarkers

Cell lines are a useful model for EV studies, because a large number of cells can be cultured for the purposes of EV isolation. This raises an issue with larger volumes of cell culture supernatant, which can be overcome using a concentration step, usually with centrifugation, in order to then load SEC columns with a manageable volume. Furthermore, if EV-depleted media is used [72], and there is careful consideration of culture supplements [73], then the expression of EV related molecules can be directly linked to the cell type under investigation.

Sakha et al. [63] used SEC to investigate a previously established highly metastatic human oral squamous cell carcinoma (SCC) cell line (HOC313-LM) and found it to be superior to dUC, or the proprietary kit by Thermo Fisher (Waltham, MA, USA). Subsequently, they were able to compare this cell line with its less metastatic parental cells (HOC313-P). Microarray analysis identified a greater level of miRNA expression in the exosomal cargo, and 7 oncogenic miRNA candidates (miR-17, miR-30a-3p, miR-30a-5p, miR-92a, miR-181a, miR-342-3p, and miR-1246) were differentially expressed in both cellular and exosomal compartments. Next, they showed that miR-1246 did not affect cell growth but instead enhanced migration and invasion in HOC313-P, TSU, and HeLa cells, possibly through reducing levels of the tumour suppressive protein DENND2D. However, it should be commented that they used small nucleolar RNA U6B as an endogenous control for normalising their experiments, which can be variably expressed in cancer [74], and levels in EVs may not be stable between healthy and cancer cohorts [75]. Current guidelines suggest the use of multiple controls for normalisation, with RNAs demonstrated as suitable with preliminary studies, or alternatively using a mean expression value normalisation strategy [76].

Peacock et al. [64] looked at oropharyngeal SCC and HPV and its association with EV miRNA expression. EV RNA was sequenced, and validated by RT-qPCR, with highly expressed miRNAs further investigated for their potential functions using miRTarBase. RNA sequencing identified a fairly low number of RNAs (2–15%) aligned to miRNA sequences, with the rest found to be mRNA, rRNA, tRNA, snoRNA, snRNA, and lincRNA. Fourteen miRNAs were enriched in HPV+ cell-derived EVs, whereas 19 miRNAs were enriched in HPV− EVs. As HPV+ oral cancer has a better prognosis than HPV− disease but is rising in incidence, these candidates would benefit from further functional investigation as this may allow the development of potential theranostics [77].

#### 3.3.3. EV Functionality

##### Functional Studies Demonstrate Tumour-Derived EV Mechanisms Involved in Immunomodulation

SEC-isolated EVs have been used to demonstrate unusual properties of EVs. Indeed, ex vivo studies by the Whiteside Lab have shown that tumour-derived EVs were immunosuppressive, and induced apoptosis of activated primary CD8+ cells, whilst T-cell- or dendritic-cell-derived EVs were immunostimulatory and promoted ex vivo proliferation of resting T-cells [78].

The purity of samples prepared in this way is important, as incubation experiments may give confounding results if the protein of interest is co-isolated as a contaminant. Frozen plasma does lead to electron-microscopy evidence of vesicular damage, and string-like aggregates, which are not cost-effectively removed by exogenous enzymatic treatments [32].

Tumour-derived EVs have been strongly implicated in the maintenance of the tumour microenvironment and can be shown to act in several ways to suppress T-cells. One study investigated immunosuppression via adenosine production, which is found at high concentrations in cancer tissues and known to be a crucial mediator in the alteration of immune cell functions in cancer [79]. The study by Schuler et al. [80] was able to demonstrate that EVs can deliver enzymatically active membrane-tethered CD73 to CD4+, CD39+, and regulatory T cells (Treg) lacking this enzyme. This co-operative interaction is a novel mechanism that could explain immunosuppression in the tumour microenvironment.

Hong et al. [69] showed that AML exosomes co-incubated with human natural killer (NK) cells inhibit expression of NKG2D, a major recognition receptor involved in tumour cell recognition and HNSCC exosomes’ suppressed activation and proliferation of activated T lymphocytes by downregulating expression levels of CD69 in co-culture. This was further expanded by Ludwig et al. [81], who used densitometry analysis of Western blots for an array of apoptosis-associated protein markers to show that the cargos of exosomes from patients with active cancer were enriched in proteins such as COX-2 (cyclooxygenase-2 also known as Prostaglandin-endoperoxide synthase 2), TGFβ-LAP (TGF-β Latency Associated Peptide), PD-1, CTLA-4, and TRAIL (Tumour necrosis Factor-related apoptosis-inducing ligand). They highlighted from this a potential strategy in the development of plasma exosomes, as future cancer biomarkers could be based on functional evaluations of total exosome fractions ex vivo (i.e., using flow-based apoptosis assays after co-incubation of ex vivo EVs with normal immune cell subsets, such as CD8+ Jurkat cells).

The emerging evidence for the role of EVs in the tumour microenvironment includes effects on angiogenesis in endothelial cells. Ludwig et al. [82] used cell-line-derived EVs to stimulate human umbilical vein endothelial cells, leading to an increased formation of vascular structures both in vitro and in vivo. Longitudinal analysis of plasma EVs showed distinct alterations in their ability to alter epithelial-to-mesenchymal transition (EMT) in recipient cells, a position that was supported by the work of Theodoraki et al. who investigated time dependent changes in plasma EV functionality after photodynamic therapy in patients with HNSCC [83]. By comparing functional activity before and after treatment, these papers suggest that this was directly related to the tumour phenotype. However, it remains to be seen whether this functional activity arises from EMT-related miRNAs or proteins.

Other mechanisms of immunomodulation proposed by EVs include the vesicular protein Arginase-1 (ARG1), which regulates the availability of L-arginine, and has been strongly associated as an immunomodulator in various settings [84,85]. This was investigated by Czystowska et al. [11] using SEC and immunoaffinity to enrich for EPCAM positive EVs in patients with ovarian cancer malignant ascites, using these EVs to suppress the proliferation of CD4+ and CD8+ T-cells, compared with EVs obtained from ovarian cyst fluid.

Sjöqvist et al. [86] took an unusual approach to investigating EV function, by isolating EVs from two types of healthy cells (keratinocytes and fibroblasts) and compared their anti-proliferative effects on cancer cells, in particular human SCC. This suppression was steroid resistant and was, therefore, identified as a rationale for cell sheet therapy (i.e., autologous transplantation of epithelial cells from the oral mucosa or otherwise) in oesophageal stricture prevention.

## 4. Conclusions

In summary, SEC can isolate EVs with a purity (as measured by levels of protein co-isolated) that is superior to dUC [32] as well as precipitation methods [20]. It can be used both for cell-conditioned medium and for clinical samples, thereby allowing the investigation of protein and RNA expression, as well as functional activity. Characterisation of EVs by this method, according to MISEV guidelines [8], demonstrates that they contain EV-related surface protein markers and structural morphology, making them suitable as a source for further research.

The high purity of SEC-isolated EVs leads to a clearer “signal” than is found for many other isolation methods, particularly with clinical samples. This allows a greater detection of potential cancer biomarkers particularly in view of the increased total EV protein and RNA levels that have been consistently identified by these studies in the plasma/sera of cancer patients. Reducing highly abundant miRNAs detected in total plasma is likely to generate benefits in EV research, allowing for better biomarker identification [33]. Protein complexes may involve RNA-binding proteins, such as the Argonaute family of proteins (AGO), and SEC allows protein-enriched fractions of the same sample to be simultaneously investigated [87]. Methods that lead to high protein co-isolation may also affect the linearity of high-sensitivity assays such as flow cytometry [88].

Of particular interest is the growing use of SEC for the isolation of biologically active EVs, that remain active even after freeze/thaw cycles [89]. This may be due to the lower operator-dependence, and reduced damage due to shearing forces caused by centrifugation at high speeds [19]. Additionally, the use of precipitation methods may have negative aspects on co-incubation studies, making some kits unsuitable as an isolation method if active EVs are required [90].

It is important to note that SEC has some limitations highlighted by these studies, which include the requirement for meticulous preparation of samples prior to loading [69], a maximal loading volume of 1–1.5 mL (dependent on column sizing), requirements for a concentration step if protein concentrations are low, and standardisation of column preparation. Contamination by plasma-associated proteins and lipoproteins is well documented (see Figure 2) and will have potentially impacted all studies that did not measure these in their SEC fractions [91].

Despite the small number of studies, 39% of the Minimal Information for Studies of Extracellular Vesicles (MISEV) criteria for characterisation was incomplete, which decreased to 34% when including only papers from 2014 onwards. RNA based studies were particularly poor at adhering to MISEV, with at least two features missing from all the studies. Without each of the components, it is impossible to guarantee that any RNA biomarker can be confidently associated with EVs and not the contaminants. This is consistent with the previous EV-TRACK findings, which noted that 30% (*n* = 5/17) of studies did not fully characterise EVs used for downstream RNA analysis. No experiments implemented other recommendations, such as using enzymatic digestion (lipase, proteinase, DNase, or RNase) to compare samples for non-EV-associated carriage.

The choice of EV isolation method has a significant impact on the EV yield [42], and this remains true both within SEC-based studies and compared with other methods. As this technique gains interest, it is likely that more comparative studies will also shed light on EV alterations that may occur during centrifugation compared with the much gentler gravity-dependent process of SEC. This will also require minimum reporting of parameters (as is the case with dUC, i.e., spin speed, time, rotor used/k-factor, and machine) [16], and it is hoped that future experiments using SEC will describe the type of material used for separation, dimensions of the elution column including the void volume, and the fractions ultimately classed as EVs to improve standardisation.

There continues to be innovative methods discovered for isolation, and some of these papers have been included in this review [92,93,94,95]. These demonstrate the feasibility for the isolation of EVs based on size, particularly in potential “lab-on-chip” applications. However, at present, the published results show that complex biofluids are problematic, leading to multiple preparatory steps to reduce sample complexity and various complications such as clogging and poor flow rates.

Asymmetric field flow fractionation is one such technology, which has been recently well described [96] and can be defined also as a size based EV isolation method. In this method, the solvent flows along a narrow channel (several hundreds of μm) together with a perpendicular cross flow that is created by a semipermeable membrane. Particles diffuse along a semipermeable membrane to be eluted according to size, and this is combined with real-time measurement of size by dynamic light scattering. It requires both significant time (1–2 h per sample) and specialised equipment but has the major advantage of enabling greater flexibility in separation parameters.

It is hoped that in the near future, these minor problems with SEC will be overcome, as the EVs isolated appear to be of high purity and a potential resource for the discovery of clinically useful cancer biomarkers.

## Figures and Tables

**Figure 1 cancers-12-03156-f001:**
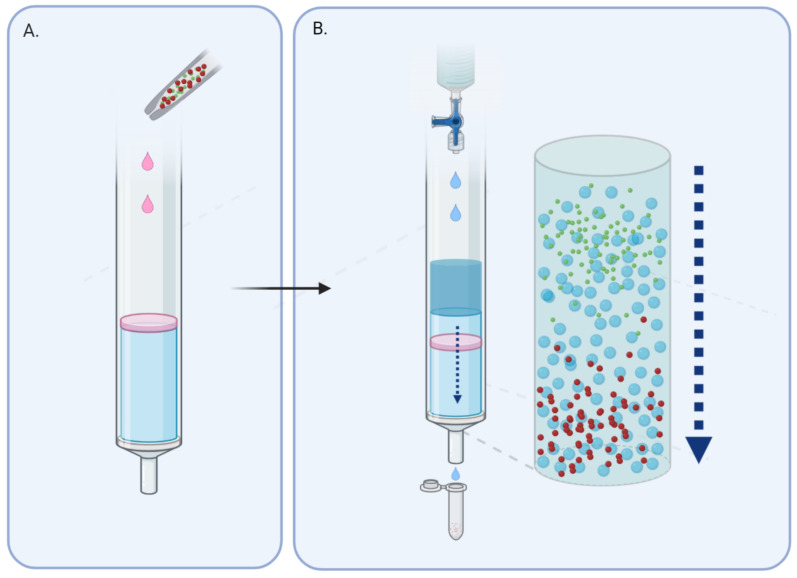
Larger particles take a more direct path through the gel and elute first. (**A**) Sample is layered onto column prepared with gel matrix. (**B**) With continuous addition of wash buffer, separation of the sample according to size occurs along the column.

**Figure 2 cancers-12-03156-f002:**
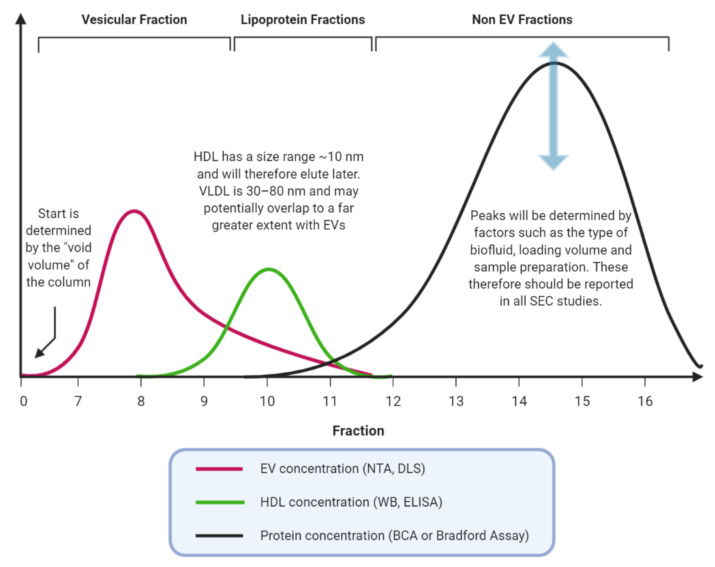
Schematic graph showing make up of consecutive fractions. EVs elute first but there is often an overlap between similarly sized particles e.g., small EVs and lipoproteins. Assessing these potential contaminants should be a mandatory feature of size-exclusion chromatography (SEC) EV studies. NTA, Nanoparticle Tracking Analysis; DLS, Dynamic Light Scattering; HDL, High-Density Lipoproteins; WB, Western Blot; ELISA, Enzyme-Linked ImmunoSorbent Assay; BCA, BicinChoninic Acid.

**Table 1 cancers-12-03156-t001:** Separation methods in brief.

Method	Frequency
Sepharose (either 2B or 4B)	13
MiniSEC (using Sepharose 2B)	10
qEV column (proprietary filtration matrix)	10
Sephacryl (S-400 or 500 HR)	2
Microfluidic Platforms	2
EVSecond (modified styrene-divinylbenzene matrix)	1
Bio-Gel A (Agarose)	1

MiniSEC, Method utilised by the Whiteside Lab (Pittsburgh, PA, USA); qEV, Proprietary columns trademarked by Izon Science (Christchurch, New Zealand); HR, High Resolution.

**Table 2 cancers-12-03156-t002:** The EV-METRIC.

**Experimental Parameters Related to EV Separation**	
Density Gradient	Type and concentrations of the gradient performed, use as validation of EVs in experiments.
EV Density	EV density should be reported if undertaken.
Ultracentrifugation Specifics	When ultracentrifugation is used, there should be a description of the *g*-force, duration and rotor type of these steps.
**Protein Analysis**	
EV-Enriched Proteins	3 or more EV-enriched proteins.
Non EV-Enriched Protein	Assessment of one or more EV-enriched protein.
Antibody Specifics	Antibodies used for immunoblotting should be reported to include clone/reference number and dilutions.
Lysate Preparation	Lysis of EVs should involve reporting of buffer composition.
**Particle Analysis**	
Qualitative and Quantitative Analysis	Implementation of both qualitative (e.g., electron microscopy) and quantitative methods (e.g., nanoparticle tracking analysis, high-resolution flow cytometry). Quantitative methods should include reporting of particle concentration.
Electron Microscopy Images	Inclusion of both wide-field and a close-up electron microscopy image.

EV, extracellular vesicle. Bold is used to mark the different table headings.

**Table 3 cancers-12-03156-t003:** Omics approaches of studies reviewed.

Study Type	Number
Proteomics	13
Biological Function	9
Feasibility of Method	11
RNA	4

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
