# Peer review of "Size-Exclusion Chromatography as a Technique for the Investigation of Novel Extracellular Vesicles in Cancer"

_cancers, 2020, doi:10.3390/cancers12113156_

Round 1

Reviewer 1 Report

In this review, the authors gathered and screened currently available data on the use of size exclusion chromatography (SEC) to isolate, purify/analyze extracellular vesicles in cancer. This collection will add a substantial contribution in the field and I therefore recommend that this review being published in it current form.

Author Response

Dear Reviewer 1,

Thank you for your input to my work "Size Exclusion Chromatography as a Technique for the Investigation of Novel Extracellular Vesicles in Cancer".

Kind regards.

Reviewer 2 Report

The manuscript submitted by Liu et al. is a detailed review on the application of SEC for the investigation of extracellular vesicles. The authors report and discuss critically the different areas of EV research in which SEC has been implemented. They give a short overview about the SEC principle and set-up. Finally, the limits and the potential of this technique for EV investigation are discussed with some very helpful advises for the interpretation of the experimental results.

The review is of significant importance for scientists working in the EV’s field. Taking into account the recent efforts for chemical modification and improvement the properties of EV, in-depth analysis of these vesicles using separation methods is receiving more attention in recent years. Therefore, I strongly recommend the publication of this review, which could be a helpful guide and motivation for using more accurate analytical techniques for EV separation.

Before publication, however, a major revision should be performed:

  1. The authors mention the use of ultracentrifugation as an additional possibility to separate EV along with SEC. Although this review is focussed on SEC, it is important to draw a short parallel to other separation techniques as it is done with UC. However, the authors do not take into account the newest developments in this field, namely the separation of EV using filed flow fractionation. There is an extensive protocol on this type of separation published in 2019 together with several recent reports.
  2. For scientists new to SEC for EV separation, it would be highly beneficial to obtain information from this review about the possible size ranges of EV’s which can be separated using SEC and how they are related to the column’s characteristic parameters. This information should be included in this review.
  3. As the authors shortly mention in the conclusions part, in SEC strong shear forces are applied. What is the impact of these forces on exosomes and are these vesicles partially modified after the SEC separation process. These are question which should be discussed.
  4. The referencing is quite confusing and should be significantly revised. The reference list includes 45 references while the citations in the text exceed this number. Thus, it is not easy to assess the quality of the bibliography.

Author Response

Dear Reviewer 2. Many thanks for your kind words and helpful comments on this manuscript. I do hope that this revised edition will have answered any issues that have arisen. Your critiques have been taken on board and I have endeavoured to improve its general readability for the research community who may wish to learn more about SEC as a general technique for cancer biomarkers.

  1. The authors mention the use of ultracentrifugation as an additional possibility to separate EV along with SEC. Although this review is focussed on SEC, it is important to draw a short parallel to other separation techniques as it is done with UC. However, the authors do not take into account the newest developments in this field, namely the separation of EV using filed flow fractionation. There is an extensive protocol on this type of separation published in 2019 together with several recent reports.

Thank you for your comments. As per your advice, I have extended my explanation on the various alternatives currently known about in my introduction so the reader is able to understand the broader picture. Included in this is reference to another paper by Taylor et al which I believe more comprehensively reviews the evidence regarding choice of isolation methods.

Assymetrical field flow fractionation is similarly a classical technique which has been of interest recently due to its use in EV isolation. The protocol you may be referring to referring to is Zhang et al https://www.nature.com/articles/s41596-019-0126-x from the Weill Cornell Medicine in New York. They have published in this area using the technique to characterise even smaller particles termed exomeres. The reported benefits of the machine are improved sensitivity over a wide size range and lack of a stationary phase which is better with viscous samples.

It can be considered a technical process which nevertheless I have now added a discussion of to the conclusion which discusses the future of size-based EV isolation as I agree this may be of general interest for future research. Unfortunately there have been some exciting papers, including involving urinary EVs using AF4 as a characterisation method for the use of SEC but this was not looked at in cancer samples and therefore outside the remit of this paper. It may be of interest to the reviewer that the authors noted that AF4 can combine isolation and characterisation but in a limited volume, and a processing time of over an hour per sample with no upscaling conceivable for the purposes of clinical diagnostics. I have therefore limited my discussion to highlighting the noted Nature Cell Bio papers and its merits.

  1. For scientists new to SEC for EV separation, it would be highly beneficial to obtain information from this review about the possible size ranges of EV’s which can be separated using SEC and how they are related to the column’s characteristic parameters. This information should be included in this review.

The major benefit of SEC for EV separation is that the size ranges of EVs is determined by the choice of stationary phase, column bed and loading sample sizes. All these characteristics combined with good characterisation of the eluted fractions will contribute to the EVs examined.

Unfortunately quantification in EV research is limited by use of NTA and DLS which have a lower size range of 10-35nm (dependent on the refractive index of the particle; highly refractive gold particles can be detected at a size of 10nm whilst protein and polydisperse samples can only be reliably detected at 35nm) https://www.sciencedirect.com/science/article/pii/S0939641116301370?via%3Dihub . Size ranges will also be determined by the stationary phase used as well as the stage of elution and this information can only be determined experimentally in each case so I would be hesitant to give more absolute values. I have however added the published data on column characteristics within my introduction to help the general reader with an interest in SEC (see Line 110).

  1. As the authors shortly mention in the conclusions part, in SEC strong shear forces are applied. What is the impact of these forces on exosomes and are these vesicles partially modified after the SEC separation process. These are question which should be discussed.

Thank you for your comments, I have clarified in the conclusions that the reverse is true, SEC in fact imparts very low shear forces on EVs as it is gravity dependent. To the authors knowledge, there are no comparative studies looking into AF4 vs SEC as the technology is still in its infancy. However it can be noted that SEC in our experience isolates EVs in a more intact form than UC and certainly precipitation kits (Gámez-Valero, A., Monguió-Tortajada, M., Carreras-Planella, L. et al. Size-Exclusion Chromatography-based isolation minimally alters Extracellular Vesicles’ characteristics compared to precipitating agents. Sci Rep 6, 33641 (2016). https://doi.org/10.1038/srep33641). “Partial modification of vesicles” may require some clarification by the reviewer as there is no current way of experimentally determining the native state of EVs without separation. One of the best ways we currently have to image EVs is cryo-EM which the above referenced paper did indeed use to great effect to demonstrate intact, clean EVs from SEC.

  1. The referencing is quite confusing and should be significantly revised. The reference list includes 45 references while the citations in the text exceed this number. Thus, it is not easy to assess the quality of the bibliography.

Thank you for spotting this major error, this was spotted by both reviewers and I have corrected this in the draft provided.

Reviewer 3 Report

The study is a review of SEC for purification of EV. The review is comprehensive and reads well. It should be of importance to the community.

  1. There is a problem with the reference list that should be corrected (45 references but numbering goes into the eighties).
  2. Typos: line 46: suggests, line 249-250: there is something missing in this sentence, line 327: a period is missing, line 336: place obtained from EVs before of cHL, line 420: has SCC been defined, 
  3. Fig 2 can be moved to supplemental materials.

Author Response

Dear Reviewer 3, many thanks for your helpful suggestions. I have amended these as follows.

  1. There is a problem with the reference list that should be corrected (45 references but numbering goes into the eighties).

Sincere apologies, I have corrected this now for the mistake which occurred during editing.

  1. Typos: line 46: suggests, line 249-250: there is something missing in this sentence, line 327: a period is missing, line 336: place obtained from EVs before of cHL, line 420: has SCC been defined, 

Thanks for noticing these mistakes, I have adjusted line 46, made line 249-250 clearer for readability, added a period to line 327, adjusted line 336 and I have made clearer that SCC is defined in line 361.

  1. Fig 2 can be moved to supplemental materials.

Thank you for your comment regarding this. I have moved it to Supplementary Figures in order to improve readability.

Round 2

Reviewer 2 Report

The authors revised carefully the manuscript. I can recommend publication without further changes.